# Discriminative Unsupervised Feature Learning with Convolutional Neural Networks

**Alexey Dosovitskiy, Jost Tobias Springenberg, Martin Riedmiller and Thomas Brox**
Department of Computer Science
University of Freiburg
79110, Freiburg im Breisgau, Germany
`{dosovits,springj,riedmiller,brox}@cs.uni-freiburg.de`

## Abstract

Current methods for training convolutional neural networks depend on large amounts of labeled samples for supervised training. In this paper we present an approach for training a convolutional neural network using only unlabeled data. We train the network to discriminate between a set of surrogate classes. Each surrogate class is formed by applying a variety of transformations to a randomly sampled 'seed' image patch. We find that this simple feature learning algorithm is surprisingly successful when applied to visual object recognition. The feature representation learned by our algorithm achieves classification results matching or outperforming the current state-of-the-art for unsupervised learning on several popular datasets (STL-10, CIFAR-10, Caltech-101).

## 1 Introduction

Convolutional neural networks (CNNs) trained via backpropagation were recently shown to perform well on image classification tasks with millions of training images and thousands of categories [1, 2]. The feature representation learned by these networks achieves state-of-the-art performance not only on the classification task for which the network was trained, but also on various other visual recognition tasks, for example: classification on Caltech-101 [2, 3], Caltech-256 [2] and the Caltech-UCSD birds dataset [3]; scene recognition on the SUN-397 database [3]; detection on the PASCAL VOC dataset [4]. This capability to generalize to new datasets makes supervised CNN training an attractive approach for generic visual feature learning.

The downside of supervised training is the need for expensive labeling, as the amount of required labeled samples grows quickly the larger the model gets. The large performance increase achieved by methods based on the work of Krizhevsky et al. [1] was, for example, only possible due to massive efforts on manually annotating millions of images. For this reason, unsupervised learning – although currently underperforming – remains an appealing paradigm, since it can make use of raw unlabeled images and videos. Furthermore, on vision tasks outside classification it is not even certain whether training based on object class labels is advantageous. For example, unsupervised feature learning is known to be beneficial for image restoration [5] and recent results show that it outperforms supervised feature learning also on descriptor matching [6].

In this work we combine the power of a discriminative objective with the major advantage of unsupervised feature learning: cheap data acquisition. We introduce a novel training procedure for convolutional neural networks that does not require any labeled data. It rather relies on an automatically generated surrogate task. The task is created by taking the idea of data augmentation – which is commonly used in supervised learning – to the extreme. Starting with trivial surrogate classes consisting of one random image patch each, we augment the data by applying a random set of transformations to each patch. Then we train a CNN to classify these surrogate classes. We refer to this method as exemplar training of convolutional neural networks (Exemplar-CNN).

The feature representation learned by Exemplar-CNN is, by construction, discriminative and invariant to typical transformations. We confirm this both theoretically and empirically, showing that this approach matches or outperforms all previous unsupervised feature learning methods on the standard image classification benchmarks STL-10, CIFAR-10, and Caltech-101.

## 1.1 Related Work

Our approach is related to a large body of work on unsupervised learning of invariant features and training of convolutional neural networks.

Convolutional training is commonly used in both supervised and unsupervised methods to utilize the invariance of image statistics to translations (e.g. LeCun et al. [7], Kavukcuoglu et al. [8], Krizhevsky et al. [1]). Similar to our approach the current surge of successful methods employing convolutional neural networks for object recognition often rely on data augmentation to generate additional training samples for their classification objective (e.g. Krizhevsky et al. [1], Zeiler and Fergus [2]). While we share the architecture (a convolutional neural network) with these approaches, our method does not rely on any labeled training data.

In unsupervised learning, several studies on learning invariant representations exist. Denoising autoencoders [9], for example, learn features that are robust to noise by trying to reconstruct data from randomly perturbed input samples. Zou et al. [10] learn invariant features from video by enforcing a temporal slowness constraint on the feature representation learned by a linear autoencoder. Sohn and Lee [11] and Hui [12] learn features invariant to local image transformations. In contrast to our discriminative approach, all these methods rely on directly modeling the input distribution and are typically hard to use for jointly training multiple layers of a CNN.

The idea of learning features that are invariant to transformations has also been explored for supervised training of neural networks. The research most similar to ours is early work on tangent propagation [13] (and the related double backpropagation [14]) which aims to learn invariance to small predefined transformations in a neural network by directly penalizing the derivative of the output with respect to the magnitude of the transformations. In contrast, our algorithm does not regularize the derivative explicitly. Thus it is less sensitive to the magnitude of the applied transformation.

This work is also loosely related to the use of unlabeled data for regularizing supervised algorithms, for example self-training [15] or entropy regularization [16]. In contrast to these semi-supervised methods, Exemplar-CNN training does not require any labeled data.

Finally, the idea of creating an auxiliary task in order to learn a good data representation was used by Ahmed et al. [17], Collobert et al. [18].

## 2 Creating Surrogate Training Data

The input to the training procedure is a set of unlabeled images, which come from roughly the same distribution as the images to which we later aim to apply the learned features. We randomly sample $N \in [50, 32000]$ patches of size $32 \times 32$ pixels from different images at varying positions and scales forming the initial training set $X = \{\mathbf{x}_1, \dots \mathbf{x}_N\}$. We are interested in patches containing objects or parts of objects, hence we sample only from regions containing considerable gradients.

We define a family of transformations $\{T_\alpha \,|\, \alpha \in \mathcal{A}\}$ parameterized by vectors $\alpha \in \mathcal{A}$, where $\mathcal{A}$ is the set of all possible parameter vectors. Each transformation $T_\alpha$ is a *composition* of elementary transformations from the following list:

- translation: vertical or horizontal translation by a distance within $0.2$ of the patch size;
- scaling: multiplication of the patch scale by a factor between $0.7$ and $1.4$;
- rotation: rotation of the image by an angle up to $20$ degrees;
- contrast 1: multiply the projection of each patch pixel onto the principal components of the set of all pixels by a factor between $0.5$ and $2$ (factors are independent for each principal component and the same for all pixels within a patch);
- contrast 2: raise saturation and value (S and V components of the HSV color representation) of all pixels to a power between $0.25$ and $4$ (same for all pixels within a patch), multiply these values by a factor between $0.7$ and $1.4$, add to them a value between $-0.1$ and $0.1$;

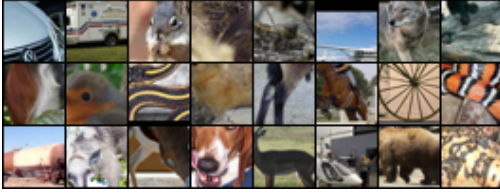 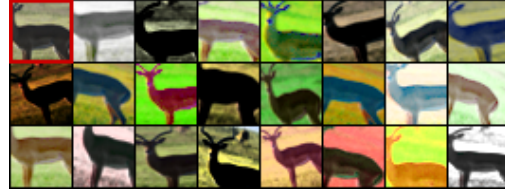

Figure 1: Exemplary patches sampled from the STL unlabeled dataset which are later augmented by various transformations to obtain surrogate data for the CNN training.

Figure 2: Several random transformations applied to one of the patches extracted from the STL unlabeled dataset. The original ('seed') patch is in the top left corner.

- color: add a value between $-0.1$ and $0.1$ to the hue (H component of the HSV color representation) of all pixels in the patch (the same value is used for all pixels within a patch).

All numerical parameters of elementary transformations, when concatenated together, form a single parameter vector $\alpha$. For each initial patch $\mathbf{x}_i \in X$ we sample $K \in [1, 300]$ random parameter vectors $\{\alpha_i^1, \ldots, \alpha_i^K\}$ and apply the corresponding transformations $\mathcal{T}_i = \{T_{\alpha_i^1}, \ldots, T_{\alpha_i^K}\}$ to the patch $\mathbf{x}_i$. This yields the set of its transformed versions $S_{\mathbf{x}_i} = \mathcal{T}_i \mathbf{x}_i = \{T\mathbf{x}_i \mid T \in \mathcal{T}_i\}$. Afterwards we subtract the mean of each pixel over the whole resulting dataset. We do not apply any other preprocessing. Exemplary patches sampled from the STL-10 unlabeled dataset are shown in Fig. 1. Examples of transformed versions of one patch are shown in Fig. 2 .

## 3   Learning Algorithm

Given the sets of transformed image patches, we declare each of these sets to be a class by assigning label $i$ to the class $S_{x_i}$. We next train a CNN to discriminate between these surrogate classes. Formally, we minimize the following loss function:

$$L(X) = \sum_{\mathbf{x}_i \in X} \sum_{T \in \mathcal{T}_i} l(i, T\mathbf{x}_i), \tag{1}$$

where $l(i, T\mathbf{x}_i)$ is the loss on the transformed sample $T\mathbf{x}_i$ with (surrogate) true label $i$. We use a CNN with a softmax output layer and optimize the multinomial negative log likelihood of the network output, hence in our case

$$l(i, T\mathbf{x}_i) = M(\mathbf{e}_i, f(T\mathbf{x}_i)),$$
$$M(\mathbf{y}, \mathbf{f}) = -\langle \mathbf{y}, \log \mathbf{f} \rangle = -\sum_k y_k \log f_k, \tag{2}$$

where $f(\cdot)$ denotes the function computing the values of the output layer of the CNN given the input data, and $\mathbf{e}_i$ is the $i$th standard basis vector. We note that in the limit of an infinite number of transformations per surrogate class, the objective function (1) takes the form

$$\widehat{L}(X) = \sum_{\mathbf{x}_i \in X} \mathbb{E}_\alpha[l(i, T_\alpha \mathbf{x}_i)], \tag{3}$$

which we shall analyze in the next section.

Intuitively, the classification problem described above serves to ensure that different input samples can be distinguished. At the same time, it enforces invariance to the specified transformations. In the following sections we provide a foundation for this intuition. We first present a formal analysis of the objective, separating it into a well defined classification problem and a regularizer that enforces invariance (resembling the analysis in Wager et al. [19]). We then discuss the derived properties of this classification problem and compare it to common practices for unsupervised feature learning.

### 3.1   Formal Analysis

We denote by $\alpha \in \mathcal{A}$ the random vector of transformation parameters, by $g(\mathbf{x})$ the vector of activations of the second-to-last layer of the network when presented the input patch $\mathbf{x}$, by $\mathbf{W}$ the matrix

of the weights of the last network layer, by $h(\mathbf{x}) = \mathbf{W}g(\mathbf{x})$ the last layer activations before applying the softmax, and by $f(\mathbf{x}) = \text{softmax}\,(h(\mathbf{x}))$ the output of the network. By plugging in the definition of the softmax activation function

$$\text{softmax}\,(\mathbf{z}) = \exp(\mathbf{z})/\|\exp(\mathbf{z})\|_1 \tag{4}$$

the objective function (3) with loss (2) takes the form

$$\sum_{\mathbf{x}_i \in X} \mathbb{E}_\alpha \left[ -\langle \mathbf{e}_i,\, h(T_\alpha \mathbf{x}_i) \rangle + \log \|\exp(h(T_\alpha \mathbf{x}_i))\|_1 \right]. \tag{5}$$

With $\widehat{\mathbf{g}}_i = \mathbb{E}_\alpha \left[ g(T_\alpha \mathbf{x}_i) \right]$ being the average feature representation of transformed versions of the image patch $\mathbf{x}_i$ we can rewrite Eq. (5) as

$$
\begin{aligned}
&\sum_{\mathbf{x}_i \in X} \left[ -\langle \mathbf{e}_i,\, \mathbf{W}\widehat{\mathbf{g}}_i \rangle + \log \|\exp(\mathbf{W}\widehat{\mathbf{g}}_i)\|_1 \right] \\
&+ \sum_{\mathbf{x}_i \in X} \left[ \mathbb{E}_\alpha \left[ \log \|\exp(h(T_\alpha \mathbf{x}_i))\|_1 \right] - \log \|\exp(\mathbf{W}\widehat{\mathbf{g}}_i)\|_1 \right].
\end{aligned} \tag{6}
$$

The first sum is the objective function of a multinomial logistic regression problem with input-target pairs $(\widehat{\mathbf{g}}_i, \mathbf{e}_i)$. This objective falls back to the transformation-free instance classification problem $\overline{L}(X) = \sum_{\mathbf{x}_i \in X} l(i, \mathbf{x}_i)$ if $g(\mathbf{x}_i) = \mathbb{E}_\alpha[g(T_\alpha \mathbf{x})]$. In general, this equality does not hold and thus the first sum enforces correct classification of the average representation $\mathbb{E}_\alpha[g(T_\alpha \mathbf{x}_i)]$ for a given input sample. For a truly invariant representation, however, the equality is achieved. Similarly, if we suppose that $T_\alpha \mathbf{x} = \mathbf{x}$ for $\alpha = 0$, that for small values of $\alpha$ the feature representation $g(T_\alpha \mathbf{x}_i)$ is approximately linear with respect to $\alpha$ and that the random variable $\alpha$ is centered, i.e. $\mathbb{E}_\alpha[\alpha] = 0$, then $\widehat{\mathbf{g}}_i = \mathbb{E}_\alpha\left[ g(T_\alpha \mathbf{x}_i) \right] \approx \mathbb{E}_\alpha\left[ g(\mathbf{x}_i) + \nabla_\alpha(g(T_\alpha \mathbf{x}_i))|_{\alpha=0}\, \alpha \right] = g(\mathbf{x}_i)$.

The second sum in Eq. (6) can be seen as a regularizer enforcing all $h(T_\alpha \mathbf{x}_i)$ to be close to their average value, i.e., the feature representation is sought to be approximately invariant to the transformations $T_\alpha$. To show this we use the convexity of the function $\log \|\exp(\cdot)\|_1$ and Jensen's inequality, which yields (proof in supplementary material)

$$\mathbb{E}_\alpha \left[ \log \|\exp(h(T_\alpha \mathbf{x}_i))\|_1 \right] - \log \|\exp(\mathbf{W}\widehat{\mathbf{g}}_i)\|_1 \geq 0. \tag{7}$$

If the feature representation is perfectly invariant, then $h(T_\alpha \mathbf{x}_i) = \mathbf{W}\widehat{\mathbf{g}}_i$ and inequality (7) turns to equality, meaning that the regularizer reaches its global minimum.

## 3.2 Conceptual Comparison to Previous Unsupervised Learning Methods

Suppose we want to unsupervisedly learn a feature representation useful for a recognition task, for example classification. The mapping from input images $\mathbf{x}$ to a feature representation $g(\mathbf{x})$ should then satisfy two requirements: (1) there must be at least one feature that is similar for images of the same category $\mathbf{y}$ (invariance); (2) there must be at least one feature that is sufficiently different for images of different categories (ability to discriminate).

Most unsupervised feature learning methods aim to learn such a representation by modeling the input distribution $p(\mathbf{x})$. This is based on the assumption that a good model of $p(\mathbf{x})$ contains information about the category distribution $p(\mathbf{y}|\mathbf{x})$. That is, if a representation is learned, from which a given sample can be reconstructed perfectly, then the representation is expected to also encode information about the category of the sample (ability to discriminate). Additionally, the learned representation should be invariant to variations in the samples that are irrelevant for the classification task, i.e., it should adhere to the manifold hypothesis (see e.g. Rifai et al. [20] for a recent discussion). Invariance is classically achieved by regularization of the latent representation, e.g., by enforcing sparsity [8] or robustness to noise [9].

In contrast, the discriminative objective in Eq. (1) does not directly model the input distribution $p(\mathbf{x})$ but learns a representation that discriminates between input samples. The representation is not required to reconstruct the input, which is unnecessary in a recognition or matching task. This leaves more degrees of freedom to model the desired variability of a sample. As shown in our analysis (see Eq. (7)), we achieve partial invariance to transformations applied during surrogate data creation by forcing the representation $g(T_\alpha \mathbf{x}_i)$ of the transformed image patch to be predictive of the surrogate label assigned to the original image patch $\mathbf{x}_i$.

It should be noted that this approach assumes that the transformations $T_\alpha$ do not change the identity of the image content. If we, for example, use a color transformation we will force the network to be invariant to this change and cannot expect the extracted features to perform well in a task relying on color information (such as differentiating black panthers from pumas)[1].

## 4   Experiments

To compare our discriminative approach to previous unsupervised feature learning methods, we report classification results on the STL-10 [21], CIFAR-10 [22] and Caltech-101 [23] datasets. Moreover, we assess the influence of the augmentation parameters on the classification performance and study the invariance properties of the network.

### 4.1   Experimental Setup

The datasets we test on differ in the number of classes (10 for CIFAR and STL, 101 for Caltech) and the number of samples per class. STL is especially well suited for unsupervised learning as it contains a large set of $100,000$ unlabeled samples. In all experiments (except for the dataset transfer experiment in the supplementary material) we extracted surrogate training data from the unlabeled subset of STL-10. When testing on CIFAR-10, we resized the images from $32 \times 32$ pixels to $64 \times 64$ pixels so that the scale of depicted objects roughly matches the two other datasets.

We worked with two network architectures. A "small" network was used to evaluate the influence of different components of the augmentation procedure on classification performance. It consists of two convolutional layers with $64$ filters each followed by a fully connected layer with $128$ neurons. This last layer is succeeded by a softmax layer, which serves as the network output. A "large" network, consisting of three convolutional layers with $64$, $128$ and $256$ filters respectively followed by a fully connected layer with $512$ neurons, was trained to compare our method to the state-of-the-art. In both models all convolutional filters are connected to a $5 \times 5$ region of their input. $2 \times 2$ max-pooling was performed after the first and second convolutional layers. Dropout [24] was applied to the fully connected layers. We trained the networks using an implementation based on *Caffe* [25]. Details on the training, the hyperparameter settings, and an analysis of the performance depending on the network architecture is provided in the supplementary material. Our code and training data are available at `http://lmb.informatik.uni-freiburg.de/resources`.

We applied the feature representation to images of arbitrary size by convolutionally computing the responses of all the network layers except the top softmax. To each feature map, we applied the pooling method that is commonly used for the respective dataset: 1) 4-quadrant max-pooling, resulting in 4 values per feature map, which is the standard procedure for STL-10 and CIFAR-10 [26, 10, 27, 12]; 2) 3-layer spatial pyramid, i.e. max-pooling over the whole image as well as within 4 quadrants and within the cells of a $4 \times 4$ grid, resulting in $1 + 4 + 16 = 21$ values per feature map, which is the standard for Caltech-101 [28, 10, 29]. Finally, we trained a linear support vector machine (SVM) on the pooled features.

On all datasets we used the standard training and test protocols. On STL-10 the SVM was trained on 10 pre-defined folds of the training data. We report the mean and standard deviation achieved on the fixed test set. For CIFAR-10 we report two results: (1) training the SVM on the whole CIFAR-10 training set ('CIFAR-10'); (2) the average over 10 random selections of 400 training samples per class ('CIFAR-10(400)'). For Caltech-101 we followed the usual protocol of selecting 30 random samples per class for training and not more than 50 samples per class for testing. This was repeated 10 times.

### 4.2   Classification Results

In Table 1 we compare Exemplar-CNN to several unsupervised feature learning methods, including the current state-of-the-art on each dataset. We also list the state-of-the-art for supervised learning (which is not directly comparable). Additionally we show the dimensionality of the feature vectors

Table 1: Classification accuracies on several datasets (in percent). † Average per-class accuracy[2] $78.0\% \pm 0.4\%$. ‡ Average per-class accuracy $84.4\% \pm 0.6\%$.

| Algorithm | STL-10 | CIFAR-10(400) | CIFAR-10 | Caltech-101 | #features |
|---|---|---|---|---|---|
| Convolutional K-means Network [26] | $60.1 \pm 1$ | $70.7 \pm 0.7$ | **82.0** | — | 8000 |
| Multi-way local pooling [28] | — | — | — | $77.3 \pm 0.6$ | $1024 \times 64$ |
| Slowness on videos [10] | 61.0 | — | — | 74.6 | 556 |
| Hierarchical Matching Pursuit (HMP) [27] | $64.5 \pm 1$ | — | — | — | 1000 |
| Multipath HMP [29] | — | — | — | $82.5 \pm 0.5$ | 5000 |
| View-Invariant K-means [12] | 63.7 | $72.6 \pm 0.7$ | 81.9 | — | 6400 |
| Exemplar-CNN (64c5-64c5-128f) | $67.1 \pm 0.3$ | $69.7 \pm 0.3$ | 75.7 | $79.8 \pm 0.5$† | 256 |
| Exemplar-CNN (64c5-128c5-256c5-512f) | $\mathbf{72.8 \pm 0.4}$ | $\mathbf{75.3 \pm 0.2}$ | **82.0** | $\mathbf{85.5 \pm 0.4}$‡ | 960 |
| Supervised state of the art | 70.1[30] | — | 91.2 [31] | 91.44 [32] | — |

produced by each method before final pooling. The small network was trained on 8000 surrogate classes containing 150 samples each and the large one on 16000 classes with 100 samples each.

The features extracted from the larger network match or outperform the best prior result on all datasets. This is despite the fact that the dimensionality of the feature vector is smaller than that of most other approaches and that the networks are trained on the STL-10 unlabeled dataset (i.e. they are used in a transfer learning manner when applied to CIFAR-10 and Caltech 101). The increase in performance is especially pronounced when only few labeled samples are available for training the SVM (as is the case for all the datasets except full CIFAR-10). This is in agreement with previous evidence that with increasing feature vector dimensionality and number of labeled samples, training an SVM becomes less dependent on the quality of the features [26, 12]. Remarkably, on STL-10 we achieve an accuracy of $72.8\%$, which is a large improvement over all previously reported results.

### 4.3 Detailed Analysis

We performed additional experiments (using the "small" network) to study the effect of three design choices in Exemplar-CNN training and validate the invariance properties of the learned features. Experiments on sampling 'seed' patches from different datasets can be found in the supplementary.

#### 4.3.1 Number of Surrogate Classes

We varied the number $N$ of surrogate classes between 50 and 32000. As a sanity check, we also tried classification with random filters. The results are shown in Fig. 3.

Clearly, the classification accuracy increases with the number of surrogate classes until it reaches an optimum at about 8000 surrogate classes after which it did not change or even decreased. This is to be expected: the larger the number of surrogate classes, the more likely it is to draw very similar or even identical samples, which are hard or impossible to discriminate. Few such cases are not detrimental to the classification performance, but as soon as such collisions dominate the set of surrogate labels, the discriminative loss is no longer reasonable and training the network to the surrogate task no longer succeeds. To check the validity of this explanation we also plot in Fig. 3 the classification error on the validation set (taken from the surrogate data) computed after training the network. It rapidly grows as the number of surrogate classes increases. We also observed that the optimal number of surrogate classes increases with the size of the network (not shown in the figure), but eventually saturates. This demonstrates the main limitation of our approach to randomly sample 'seed' patches: it does not scale to arbitrarily large amounts of unlabeled data. However, we do not see this as a fundamental restriction and discuss possible solutions in Section 5 .

#### 4.3.2 Number of Samples per Surrogate Class

Fig. 4 shows the classification accuracy when the number $K$ of training samples per surrogate class varies between 1 and 300. The performance improves with more samples per surrogate class and

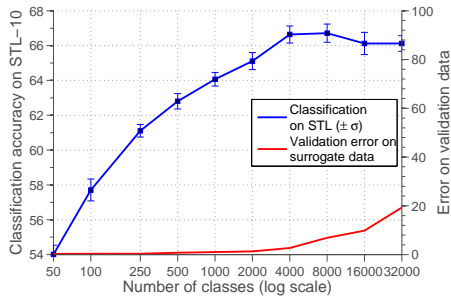

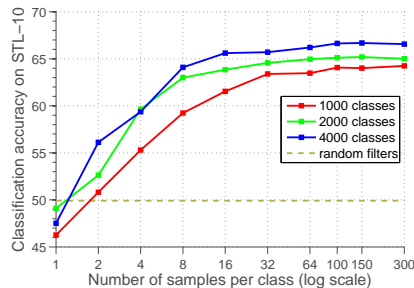

Figure 3: Influence of the number of surrogate training classes. The validation error on the surrogate data is shown in red. Note the different y-axes for the two curves.

Figure 4: Classification performance on STL for different numbers of samples per class. Random filters can be seen as '0 samples per class'.

saturates at around 100 samples. This indicates that this amount is sufficient to approximate the formal objective from Eq. (3), hence further increasing the number of samples does not significantly change the optimization problem. On the other hand, if the number of samples is too small, there is insufficient data to learn the desired invariance properties.

### 4.3.3 Types of Transformations

We varied the transformations used for creating the surrogate data to analyze their influence on the final classification performance. The set of 'seed' patches was fixed. The result is shown in Fig. 5. The value '0' corresponds to applying random compositions of all elementary transformations: scaling, rotation, translation, color variation, and contrast variation. Different columns of the plot show the difference in classification accuracy as we discarded some types of elementary transformations.

Several tendencies can be observed. First, rotation and scaling have only a minor impact on the performance, while translations, color variations and contrast variations are significantly more important. Secondly, the results on STL-

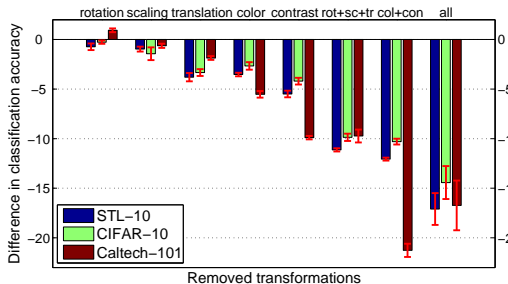

Figure 5: Influence of removing groups of transformations during generation of the surrogate training data. Baseline ('0' value) is applying all transformations. Each group of three bars corresponds to removing some of the transformations.

10 and CIFAR-10 consistently show that spatial invariance and color-contrast invariance are approximately of equal importance for the classification performance. This indicates that variations in color and contrast, though often neglected, may also improve performance in a supervised learning scenario. Thirdly, on Caltech-101 color and contrast transformations are much more important compared to spatial transformations than on the two other datasets. This is not surprising, since Caltech-101 images are often well aligned, and this dataset bias makes spatial invariance less useful.

### 4.3.4 Invariance Properties of the Learned Representation

In a final experiment, we analyzed to which extent the representation learned by the network is invariant to the transformations applied during training. We randomly sampled 500 images from the STL-10 test set and applied a range of transformations (translation, rotation, contrast, color) to each image. To avoid empty regions beyond the image boundaries when applying spatial transformations, we cropped the central $64 \times 64$ pixel sub-patch from each $96 \times 96$ pixel image. We then applied two measures of invariance to these patches.

First, as an explicit measure of invariance, we calculated the normalized Euclidean distance between normalized feature vectors of the original image patch and the transformed one [10] (see the supplementary material for details). The downside of this approach is that the distance between extracted features does not take into account how informative and discriminative they are. We there-

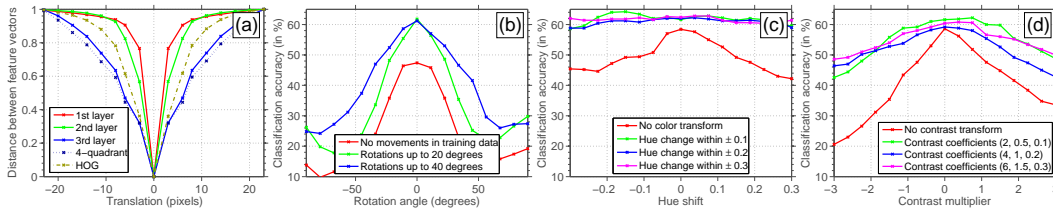

Figure 6: Invariance properties of the feature representation learned by Exemplar-CNN. (a): Normalized Euclidean distance between feature vectors of the original and the translated image patches vs. the magnitude of the translation, (b)-(d): classification performance on transformed image patches vs. the magnitude of the transformation for various magnitudes of transformations applied for creating surrogate data. (b): rotation, (c): additive color change, (d): multiplicative contrast change.

fore evaluated a second measure – classification performance depending on the magnitude of the transformation applied to the classified patches – which does not come with this problem. To compute the classification accuracy, we trained an SVM on the central $64 \times 64$ pixel patches from one fold of the STL-10 training set and measured classification performance on all transformed versions of $500$ samples from the test set.

The results of both experiments are shown in Fig. 6 . Due to space restrictions we show only few representative plots. Overall the experiment empirically confirms that the Exemplar-CNN objective leads to learning invariant features. Features in the third layer and the final pooled feature representation compare favorably to a HOG baseline (Fig. 6 (a)). Furthermore, adding stronger transformations in the surrogate training data leads to more invariant classification with respect to these transformations (Fig. 6 (b)-(d)). However, adding too much contrast variation may deteriorate classification performance (Fig. 6 (d)). One possible reason is that level of contrast can be a useful feature: for example, strong edges in an image are usually more important than weak ones.

## 5   Discussion

We have proposed a discriminative objective for unsupervised feature learning by training a CNN without class labels. The core idea is to generate a set of surrogate labels via data augmentation. The features learned by the network yield a large improvement in classification accuracy compared to features obtained with previous unsupervised methods. These results strongly indicate that a discriminative objective is superior to objectives previously used for unsupervised feature learning.

One potential shortcoming of the proposed method is that in its current state it does not scale to arbitrarily large datasets. Two probable reasons for this are that (1) as the number of surrogate classes grows larger, many of them become similar, which contradicts the discriminative objective, and (2) the surrogate task we use is relatively simple and does not allow the network to learn invariance to complex variations, such as 3D viewpoint changes or inter-instance variation. We hypothesize that the presented approach could learn more powerful higher-level features, if the surrogate data were more diverse. This could be achieved by using additional weak supervision, for example, by means of video or a small number of labeled samples. Another possible way of obtaining richer surrogate training data and at the same time avoiding similar surrogate classes would be (unsupervised) merging of similar surrogate classes. We see these as interesting directions for future work.

## Acknowledgements

We acknowledge funding by the ERC Starting Grant VideoLearn (279401); the work was also partly supported by the BrainLinks-BrainTools Cluster of Excellence funded by the German Research Foundation (DFG, grant number EXC 1086).

## Footnotes

[1]Such cases could be covered either by careful selection of applied transformations or by combining features from multiple networks trained with different sets of transformations and letting the final classifier choose which features to use.

[2] On Caltech-101 one can either measure average accuracy over all samples (average overall accuracy) or calculate the accuracy for each class and then average these values (average per-class accuracy). These differ, as some classes contain fewer than 50 test samples. Most researchers in ML use average overall accuracy.

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
