[Supplementary Material]

# Supplementary Material for "Discriminative Unsupervised Feature Learning with Convolutional Neural Networks"

**Alexey Dosovitskiy, Jost Tobias Springenberg and Thomas Brox**
Department of Computer Science
University of Freiburg
79110, Freiburg im Breisgau, Germany
{dosovits,springj,brox}@cs.uni-freiburg.de

## 1   Formal Analysis

In this section we present the proofs for the formal analysis from Section 3.1 of our paper.

**Proposition 1** *The function*

$$Z(\mathbf{x}) = \log \| \exp(\mathbf{x}) \|_1, \ \mathbf{x} \in \mathbb{R}^n$$

*is convex. Moreover, for any $\mathbf{x} \in \mathbb{R}^n$ the kernel of its Hessian matrix $\nabla^2 Z(\mathbf{x})$ is given by $span\,(\mathbf{1})$*

**Proof**  Since

$$Z(\mathbf{x}) = \log \| \exp(\mathbf{x}) \|_1 = \log \sum_{i=1}^{n} \exp(x_i) \tag{1}$$

we need to prove the convexity of the log-sum-exp function. The Hessian $\nabla^2$ of this function is given as

$$\nabla^2 Z(\mathbf{x}) = \frac{1}{(\mathbf{1}^T \mathbf{u})^2}((\mathbf{1}^T \mathbf{u})\, diag\,(\mathbf{u}) - \mathbf{u}\mathbf{u}^T), \tag{2}$$

with $\mathbf{u} = exp(\mathbf{x})$ and $\mathbf{1} \in \mathbb{R}^n$ being a vector of ones. To show the convexity we must prove that $\mathbf{z}^T \nabla^2 Z(\mathbf{x})\mathbf{z} \geq 0$ for all $\mathbf{x}, \mathbf{z} \in \mathbb{R}^n$. From (2) we get

$$\mathbf{z}^T \nabla^2 Z(\mathbf{x})\, \mathbf{z} = \frac{1}{(\mathbf{1}^T \mathbf{u})^2}((\mathbf{1}^T \mathbf{u})\, \mathbf{z}^T diag\,(\mathbf{u})\, \mathbf{z} - \mathbf{z}^T \mathbf{u}\mathbf{u}^T \mathbf{z})$$

$$= \frac{(\sum_{k=1}^{n} u_k z_k^2)(\sum_{k=1}^{n} u_k) - (\sum_{k=1}^{n} u_k z_k)^2}{(\sum_{k=1}^{n} u_k)^2} \geq 0 \tag{3}$$

since $(\sum_{k=1}^{n} u_k)^2 \geq 0$ and $(\sum_{k=1}^{n} z_k u_k)^2 \leq (\sum_{k=1}^{n} u_k z_k^2)(\sum_{k=1}^{n} u_k)$ due to the Cauchy-Schwarz inequality.

Inequality (3) only turns to equality if

$$\sqrt{u_k} z_k = c\sqrt{u_k}, \tag{4}$$

where the constant $c$ does not depend on $k$. This immediately gives $\mathbf{z} = c\mathbf{1}$, which proves the second statement of the proposition.

**Proposition 2** *Let $\alpha \in \mathcal{A}$ be a random vector with values in a bounded set $\mathcal{A} \subset \mathbb{R}^k$. Let $\mathbf{x}(\cdot)\colon \mathcal{A} \to \mathbb{R}^n$ be a continuous function. Then inequality*

$$\mathbb{E}_\alpha \left[ \log \| \exp(\mathbf{x}(\alpha)) \|_1 \right] - \log \| \exp(\mathbb{E}_\alpha[\mathbf{x}(\alpha)]) \|_1 \geq 0 \tag{5}$$

*holds and only turns to equality if for all $\alpha_1, \alpha_2 \in \mathcal{A}$: $(\mathbf{x}(\alpha_1) - \mathbf{x}(\alpha_2)) \in span\,(\mathbf{1})$.*

**Proof** Inequality (5) immediately follows from convexity of the function $\log \| \exp(\cdot) \|_1$ and Jensen's inequality.

Jensen's inequality only turns to equality if the function it is applied to is affine-linear on the convex hull of the integration region. In particular this implies

$$(\mathbf{x}(\alpha_1) - \mathbf{x}(\alpha_2))^T \nabla^2 Z(\mathbf{x}(\alpha_1)) (\mathbf{x}(\alpha_1) - \mathbf{x}(\alpha_2)) = 0 \qquad (6)$$

for all $\alpha_1, \alpha_2 \in \mathcal{A}$. The second statement of Proposition 1 thus immediately gives $\mathbf{x}(\alpha_1) - \mathbf{x}(\alpha_2) = c\mathbf{1}$, Q.E.D.

## 2 Details on Training Procedure

We describe here in detail which network architectures we tried and explain the network training procedure.

### 2.1 Network Architecture

We tested various network architectures in combination with our training procedure. They are coded as follows: NcF stands for a convolutional layer with $N$ filters of size $F \times F$ pixels, Nf stands for a fully connected layer with $N$ neurons. For example, 64c5-64c5-128f denotes a network with two convolutional layers containing 64 filters spanning $5 \times 5$ pixels each followed by a fully connected layer with 128 neurons. The last specified layer is always succeeded by a softmax layer, which serves as the network output. We applied $2 \times 2$ max-pooling to the outputs of the first and second convolutional layers.

As stated in the paper we used a 64c5-64c5-128f architecture in our experiments to evaluate the influence of different components of the augmentation procedure (we refer to this architecture as the 'small' network). A large network, coded as 64c5-128c5-256c5-512f, was then used to achieve better classification performance.

All considered networks contained rectified linear units in each layer but the softmax layer. Dropout was applied to the fully connected layer.

### 2.2 Training the Networks

We adopted the common practice of training the network with stochastic gradient descent with a fixed momentum of $0.9$. We started with a learning rate of $0.01$ and gradually decreased the learning rate during training. That is, we trained until there was no improvement in validation error, then decreased the learning rate by a factor of $3$, and repeated this procedure until convergence.

## 3 Experiments

We report here two additional experiments studying influence of different aspects of the algorithm on the quality of the learned features. We also give the details on how we measure invariance of feature representations in Section 4.3.4 of the paper.

### 3.1 Influence of the Network Architecture on Classification Performance

We perform an additional experiment to evaluate the influence of the network architecture on classification performance. The results of this experiment are shown in Table 1. All networks were trained using a surrogate training set containing either 8000 classes with 150 samples each or 16000 classes with 100 samples each (for larger networks). We vary the number of layers, layer sizes and filter sizes. Classification accuracy generally improves with the network size indicating that our classification problem scales well to relatively large networks without overfitting.

### 3.2 Influence of the Dataset

We applied our feature learning algorithm to images sampled from three datasets – STL-10 unlabeled dataset, CIFAR-10 and Caltech-101 – and evaluated the performance of the learned feature

Table 1: Classification accuracy depending on the network architecture. The name coding is as follows: NcF stands for a convolutional layer with $N$ filters of size $F \times F$ pixels, Nf stands for a fully connected layer with $N$ neurons. For example, 64c5-64c5-128f denotes a network with two convolutional layers containing 64 filters spanning $5 \times 5$ pixels each followed by a fully connected layer with 128 neurons. We also show the number of surrogate classes used for training each network.

| Architecture | #classes | STL-10 | CIFAR-10(400) | CIFAR-10 | Caltech-101 |
|---|---|---|---|---|---|
| 32c5-32c5-64f | 8000 | $63.8 \pm 0.4$ | $66.1 \pm 0.4$ | 71.3 | $78.2 \pm 0.6$ |
| 64c5-64c5-128f | 8000 | $67.1 \pm 0.3$ | $69.7 \pm 0.3$ | 75.7 | $79.8 \pm 0.5$ |
| 64c7-64c5-128f | 8000 | $66.3 \pm 0.4$ | $69.5 \pm 0.3$ | 75.0 | $79.4 \pm 0.7$ |
| 64c5-64c5-64c5-128f | 8000 | $68.5 \pm 0.3$ | $70.9 \pm 0.3$ | 77.0 | $82.2 \pm 0.7$ |
| 64c5-64c5-64c5-64c5-128f | 8000 | $64.7 \pm 0.5$ | $67.5 \pm 0.3$ | 75.2 | $75.7 \pm 0.4$ |
| 128c5-64c5-128f | 8000 | $67.2 \pm 0.4$ | $69.9 \pm 0.2$ | 76.1 | $80.1 \pm 0.5$ |
| 64c5-256c5-128f | 8000 | $69.2 \pm 0.3$ | $71.7 \pm 0.3$ | 77.9 | $81.6 \pm 0.5$ |
| 64c5-64c5-512f | 8000 | $69.0 \pm 0.4$ | $71.7 \pm 0.2$ | 79.3 | $82.9 \pm 0.4$ |
| 128c5-256c5-512f | 8000 | $71.2 \pm 0.3$ | $73.9 \pm 0.3$ | 81.5 | $84.3 \pm 0.6$ |
| 128c5-256c5-512f | 16000 | $71.9 \pm 0.3$ | $74.3 \pm 0.3$ | 81.4 | $84.6 \pm 0.6$ |
| 64c5-128c5-256c5-512f | 16000 | $72.8 \pm 0.4$ | $75.3 \pm 0.3$ | 82.0 | $85.5 \pm 0.4$ |

Table 2: Dependence of classification performance (in %) on the training and testing datasets. Each column corresponds to different test data, each row to different training data (i.e. source of seed patches). We used the "small" network (64c5-64c5-128f) for this experiment.

| | TESTING | | |
|---|---|---|---|
| TRAINING | STL-10 | CIFAR-10(400) | CALTECH-101 |
| STL-10 | $\mathbf{67.1 \pm 0.3}$ | $69.7 \pm 0.3$ | $79.8 \pm 0.5$ |
| CIFAR-10 | $64.5 \pm 0.4$ | $\mathbf{70.3 \pm 0.4}$ | $77.8 \pm 0.6$ |
| CALTECH-101 | $66.2 \pm 0.4$ | $69.5 \pm 0.2$ | $\mathbf{80.0 \pm 0.5}$ |

representations on classification tasks on these datasets. We used the "small" network (64c5-64c5-128f) for this experiment.

We show first layer filters learned from the three datasets in Fig. 1. Note how filters qualitatively differ depending on the dataset they were trained on.

Classification results are shown in Table 2. The best classification results for each dataset are obtained when training on the patches extracted from the dataset itself. However, the difference is not drastic, indicating that the learned features generalize well to other datasets.

### 3.3 Details of computing the measure of invariance

We now explain in detail and motivate computation of the normalized Euclidean distance used as a measure of invariance in the paper.

First we compute feature vectors of all image patches and their transformed versions. We next normalize each feature vector to unit Euclidean norm and compute Euclidean distances between each original patch and all of its transformed versions. For each transformation and magnitude we average these distances over all patches. Finally, we divide the resulting curves by their maximal values (typically it is the value for the maximum magnitude of the transformation).

The normalizations are performed to compensate for possibly different scales of different features. Normalizing feature vectors to unit length ensures that the values are in the same range for different features. The final normalization of the curves by the maximal value allows to compensate for different variation of different features: as an extreme, a constant feature would be considered perfectly invariant without this normalization, which is certainly not desirable.

The resulting curves show how quickly the feature representation changes when an image is transformed more and more. A representation for which the curve steeply goes up and then remains constant cannot be considered invariant to the transformation: the feature vector of the transformed patch becomes completely uncorrelated with the original feature vector even for small magnitudes of the transformation. On the other hand, if the curve grows gradually, this indicates that the feature representation changes slowly when the transformation is applied, meaning invariance or, rather, covariance of the representation.

Figure 1: Filters learned by first layers of 64c5-64c5-128f networks when training on surrogate data from various dataset. Top – from STL-10, middle – CIFAR-10, bottom – Caltech-101.