[Reviews · NeurIPS 2014]

Submitted by Assigned_Reviewer_1

This paper described a novel method to train convolutional neural networks (CNNs) in an unsupervised fashion such that the model still learns to be invariant to common transformations.

The method proposed by the authors is simple and, some extent, elegant. The idea is to simply train the CNN to distinguish image patches and their transformations from other image patches and their transformations.

This approach allows the model to learn to be invariant to common transformation. However, as the authors mention, the method is vulnerable to "collisions" where distinct image patches -- that the model will try to distinguish -- share the same content.

The method is novel and, while simple, isn't necessarily obvious (at least not to me). It's a clever idea and I think, with some minor changes (see below), it deserves to be published.

The author also present theoretical result that the authors claim support the argument that the model learning invariant features. I did not find this analysis particularly compelling as it seems to basically just state that the if the model were to find a perfect invariant feature representation, then it would achieve the global optimum of the objective function. This seems obvious from the definition of the objective function. Perhaps I've missed something more subtle.

The empirical exploration of the proposed approach is relatively thorough and insightful, including experiments carefully exploring the potential vulnerability described above as the number of surrogate "classes" (or base training patches) increases. The experimental results also include an evaluation of the invariance properties of the learned representation. On
the empirical side, this work is relatively solid and mature.

My only quibble is that in presenting the state-of-the-art results for STL-10, CIFAR-10 and Caltech-101, the authors neglect to provide a comparison with the actual state-of-the-art (particularly for CIFAR-10). They state that they are not comparing to standard discriminatively-trained CNN models -- the motivation for this line in the sand seems arbitrary and frankly a bit self-serving. The problem is that the presentation in the paper clearly leaves the reader with the impression that this method approaches the state-of-the-art for all these datasets. This is clearly not the case and it isn't acceptable to leave that impression with the reader. I ask simply that the authors provide the true state-of-the-art for each of these datasets, with and without dataset transformations.
Summary: A good paper about a simple, but clever idea well explored empirically.

Submitted by Assigned_Reviewer_31

This paper presents a new approach to discriminative unsupervised feature learning. Given an unlabelled set of images, 'seed' patches are extracted which act as surrogate classes for training. For each of these seed images, a variety of transformations is applied to generate training examples within the seed classes. A convolutional network is trained on top of the patches to predict the seed classes. After training, feature are extracted by forward propagating a new image, extracting the feature maps at each layer and pooling over them. Experimentation is performed on several standard image classification benchmarks, with additional experimentation to determine the effectiveness of transformations and invariance properties of the learned features.

The proposed approach is simple and demonstrates strong performance on each of the classification tasks. The additional analysis, particularly the effects of removing transformations, was most welcome. The paper is clearly written for the most part with enough information given for the reader to reproduce the results.

Detailed comments / questions:

- Can you go into more detail about patch extraction? Are you just sampling a large number of patches, computing the gradients for each one, and choosing the top-N?

- You should mention that the STL-10 dataset uses 96x96 images earlier in the paper, in order to give context for the choice of 32x32 patch size.

- Was the 32x32 patch size arbitrary chosen? Do you have any intuition on how to select patch sizes as a function of the image sizes?

- section 4.1: A picture or two, perhaps one illustrating the convnet during surrogate training and another illustrating feature extraction would be really helpful. The description is fairly clear for a reader familiar with this line of work but may be challenging to understand for those that are not.

- From a practical point of view, given a dataset with a small amount of labels, one of the simplest things to try is feeding the images through a pre-trained convolutional net on ImageNet and training a linear classifer on top of these features. I think it would beneficial to include these results in table 1, even through they fall under a different category of algorithms (you could include this in a separate block of the table). It would be interesting to see how your results compare (at least on Caltech 101, your results are on par to DeCAF)

- How much does validation on the surrogate task act as a proxy for validation on the classification task? Does the network with the best validation performance on the surrogate task also have the best validation performance on the classification task? Figure 3 hints that this might be true. Table 1 in the supplementary material presents classification accuracies with several networks. I would highly recommended including (maybe in a separate table) both the surrogate validation scores as well as the classification validation scores for each of these tasks, to see how well they correspond.

- Supplementary, table 2: How come the diagonal is different than the results in table 1 with the same network?

- To my best knowledge, the best published result on STL-10 is 70.1% +- 0.6% from "Multi-Task Bayesian Optimization" (Swersky et al, NIPS 2013). They achieved their results by first extracting k-means feature maps and training a convolutional network on top of these. By fixing the k-means features, the network is much less prone to overfitting. A strong baseline that I'd recommend the authors try, if time permits, is to use the same approach they do but include each of the transformations you use in the paper. This give a relatively straightforward way of running convnets on small labelled datasets which I suspect would be very competitive with your approach.
Summary: In summary, this is a nice paper with a simple algorithm that gets very good results on standard benchmarks. I suspect the generality of this method should lead to several interesting research directions. For these reasons, I recommend acceptance.

Submitted by Assigned_Reviewer_41

In this paper, the authors propose an interesting idea for learning invariances, in the unsupervised learning setting for image classification problems. The key idea is to create surrogate classes consisting of groups of transformations of randomly selected patches. The transformations are 2D functions such as rotations, scaling, translations and color shifts. The formal analysis is interesting and sheds further light on why this approach could work.

The paper is well written and easy to follow. The experiments cover a number of aspects of the problem, including number of transformations and size of network.

One drawback of the approach in the paper seems to be that two randomly selected patches which are very similar in content will be "forced apart" since they will be considered different surrogate classes. The authors implicitly sidestep this by using only a maximum of 32000 patches from large datasets. However, using very few surrogate patches may impede generalization in case of larger datasets. Do the authors have any comments on this?

Another experiment which would be nice is to have used the same set of surrogate classes for unsupervised learning and then training dataset-specific classifiers. For example, since STL-10 and CIFAR-10 are closely related, why did the authors need to retrain on each dataset?

Finally, experiments on varying patch size would also have been interesting.

Summary: In this paper, the authors propose an interesting idea for learning invariances, in the unsupervised learning setting for image classification problems. The key idea is to create surrogate classes consisting of groups of transformations of randomly selected patches. The paper's idea is novel and interesting for unsupervised learning. It is well written and easy to follow.
Author Feedback
Author rebuttal: We first want to thank all reviewers for their thorough reviews and insightful remarks.

R1: Assigned_Reviewer_1
R2: Assigned_Reviewer_31
R3: Assigned_Reviewer_41

R1+R2:
------
We excluded results of supervised methods in Table 1 to avoid confusion between supervised and unsupervised feature learning. We agree that it can be interesting to also compare to the absolute state-of-the-art on these datasets, which is obtained with supervised feature learning methods, and acknowledge that omitting these results could potentially be misleading to the reader. We will include these numbers in Table 1 in the final paper.

R2:
------
[Details on patch extraction]
Rather than taking the top N patches, we sampled patches from STL-10 at random with probability proportional to the squared Euclidean norm of their gradients. In our experiments we did not see a large difference compared to uniform sampling, since only few homogeneous patches exist in the STL unlabeled dataset. For other datasets, this might be more relevant.

[How much does validation on the surrogate task act as a proxy for validation on the classification task?]
Assessing the correlation between validation error on the surrogate task and performance on the classification task is not trivial, as it heavily depends on the way the surrogate task is set up. One could, for example, design a surrogate task in which the classes are formed by applying no transformations at all. Then with a large enough network a very low validation error can be achieved, but the network does not generalize to the classification task. If space permits, we will add a short paragraph on this issue.

[Swersky et al.]
We agree that Swersky et al. might give good results when used together with the augmentation. We are curious ourselves how well various supervised methods will perform when combined with these augmentations and will investigate this in the future. However, we believe that a proper investigation on this is out of the scope of this paper.

[Supplementary, table 2: How come the diagonal is different than the results in table 1 with the same network?]
Thank you for spotting this error in the supplementary material. The results in Table 2 are from a previous run with a different surrogate dataset, which we forgot to update. We will correct this. Please note, however, that the results from Table 1 in the paper should correspond to the first row in Table 2 of the supplementary and not to the diagonal, as for the main experiments we only used data from STL-10 for training the network.

R2+R3:
------
[32x32 patch size]
The patch size was chosen such that: (1) the applied transformations that introduce movements (rotations, large translations) would result in transformed patches that stay well within the image boundaries of the STL data; (2) the patch size roughly matches the size of the architecture. We have not yet carried out an in-depth analysis of the effect of this choice and believe its choice to be dependent on the network architecture. Preliminary tests with slightly larger patch sizes did not result in significantly different performance.

R3:
------
[One drawback of the approach seems to be that two randomly selected patches which are very similar in content will be "forced apart".]
We are aware of this drawback and already discussed this issue and potential solutions in the last section of the paper. This is a good direction for future work. We believe that especially the idea of merging similar surrogate classes during training has great potential for mitigating this drawback.

[Another nice experiment is to use the same set of surrogate classes for unsupervised learning and then training dataset-specific classifiers.]
It seems our explanation of the training procedure has caused a misunderstanding. This is exactly what we did! All experiments in the paper were performed using surrogate tasks created from the STL-10 unlabeled dataset. The labeled data from the target dataset is only used to train the SVM on the representation extracted using our network. An additional experiment on the influence of the data used for the surrogate task is in the supplementary material.